# Harmonic radar tracking of individual melon flies, *Zeugodacus cucurbitae*, in Hawaii: Determining movement parameters in cage and field settings

**Nicole D. Miller**[1], **Theodore J. Yoder**[1], **Nicholas C. Manoukis**[2], **Lori A. F. N. Carvalho**[2], **Matthew S. Siderhurst**[1] *

**1** Department of Chemistry, Eastern Mennonite University, Harrisonburg, VA, United States of America,
**2** Daniel K. Inouye US Pacific Basin Agricultural Research Center, United States Department of Agriculture, Agricultural Research Service, Hilo, HI, United States of America

* matthew.siderhurst@emu.edu

**Data Availability Statement:** All relevant data are within the manuscript and its Supporting Information files.

## Abstract

Tephritid fruit flies, such as the melon fly, *Zeugodacus cucurbitae*, are major horticultural pests worldwide and pose invasion risks due primarily to international trade. Determining movement parameters for fruit flies is critical to effective surveillance and control strategies, from setting quarantine boundaries after incursions to development of agent-based models for management. While mark-release-recapture, flight mills, and visual observations have been used to study tephritid movement, none of these techniques give a full picture of fruit fly movement in nature. Tracking tagged flies offers an alternative method which has the potential to observe individual fly movements in the field, mirroring studies conducted by ecologists on larger animals. In this study, harmonic radar (HR) tags were fabricated using superelastic nitinol wire which is light (tags weighed less than 1 mg), flexible, and does not tangle. Flight tests with wild melon flies showed no obvious adverse effects of HR tag attachment. Subsequent experiments successfully tracked HR tagged flies in large field cages, a papaya field, and open parkland. Unexpectedly, a majority of tagged flies showed strong flight directional biases with these biases varying between flies, similar to what has been observed in the migratory butterfly *Pieris brassicae*. In field cage experiments, 30 of the 36 flies observed (83%) showed directionally biased flights while similar biases were observed in roughly half the flies tracked in a papaya field. Turning angles from both cage and field experiments were non-random and indicate a strong bias toward continued "forward" movement. At least some of the observed direction bias can be explained by wind direction with a correlation observed between collective melon fly flight directions in field cage, papaya field, and open field experiments. However, individual mean flight directions coincided with the observed wind direction for only 9 out of the 25 flies in the cage experiment and half of the flies in the papaya field, suggesting wind is unlikely to be the only factor affecting flight direction. Individual flight distances (meters per flight) differed between the field cage, papaya field, and open field experiments with longer mean step-distances observed in the open field. Data on flight directionality and step-distances determined in this study might assist in

**Funding:** NCM, MSS, LAFN - This study was supported in part by ARS project 2040-22430-027-00D, United States Department of Agriculture, Agricultural Research Service, www.ars.usda.gov/ NDM, TJY - This study was supported in part by a Kauffman and Miller Research Award, Daniel B. Suter Endowment in Biology, Eastern Mennonite University, emu.edu/science-seminars/daniel-b-suter-endowment There was no additional external funding received for this study. The funders had no role in study design, data collection and analysis, decision to publish, or preparation of the manuscript.

**Competing interests:** The authors have declared that no competing interests exist.

the development of more effective control and better parametrize models of pest tephritid fruit fly movement.

## Introduction

Tephritid fruit flies are among the most important economic horticultural pests worldwide [1]. The polyphagous nature of these flies presents a wide range of threats to economically important crops [2] with species such as Medfly, *Ceratitis capitata* (Wiedemann) potentially infesting over 700 hosts [3] and Oriental fruit fly, *Bactrocera dorsalis* (Hendel), over 700 [4]. Many fruit-producing countries have imposed quarantine restrictions on the importation of products from areas with established tephritid fruit fly populations due to the potential damage introduced species might cause.

The melon fly, *Zeugodacus cucurbitae* (Coquillett), typifies damage that tephritids can cause as this pest is known to damage at least 81 host species [5]. Introduced to the Hawaiian Islands in the late 19[th] century, the melon fly is a particularly serious pest of cucurbit crops and papaya, [5] and along with medfly and the oriental fruit fly cause an estimated $US 15 million/year in direct costs to Hawaiian agriculture [6] and $US 300 million/year in lost markets [7]. Melon fly is a major quarantine pest and has been detected in California on multiple occasions [8]; their presence in Hawaii poses a serious threat to U.S mainland agriculture through accidental introduction.

To counter the threat of invasive tephritids, California, Florida, and Texas deploy trapping networks for early detection of tephritid pests as well the release of sterile males. Placement of the traps is partially affected by tephritid fly movement. When tephritids are detected, delimitation and quarantine efforts are often triggered [9, 10]. These involve applying measures such as increased trapping, insecticide application, and sterile insect technique over an area, but the size of this area is difficult to set. This is because the spread of the population will depend on multiple factors such as the length of time since the incursion and, critically, the dispersal ability of the pest fly.

Thus, understanding tephritid fruit fly movement can improve detection and control of these economically important pests. Fruit flies move to find food, oviposition sites, and protection in dense vegetation, with this movement taking place both over short distances, such as within crop fields and orchards, and long range via ship ports and human-assisted movement in urban areas.

Movement data may also provide insights into how to optimize IPM control strategies as has been done for the brown marmorated stink bug, *Halyomorpha halys* (Stål) [11] and the spotted wing drosophila, *Drosophila suzukii* (Matsumura) [12]. Additionally, fly movement data will allow better modeling of pest populations to understand potential pest distribution, quarantine deployment, optimizing trapping networks, and predicting pest outbreaks [13]. Dispersal capacity (specifically step-distance and flight directionality), spatial distribution, and density of pest insects are particularly important for improving agent-based modeling [14–17].

In areas where particular tephritids are invasive or in "fruit fly free zones" [18] measurement of fly movement in the field can be extremely helpful for improving increasingly sophisticated models aimed at improving detection, delimitation, and eradication. In recent years there have been a number of Agent-based simulations for addressing management and eradication of tephritids [14, 16, 17]. While not all of these are spatially-explicit, one of the reasons for this usually includes lack of movement parameters, such as turning angles and step-

distances, for the target species. A subclass of models focused on trapping especially benefit from realistic movement modeling [15, 19, 20], since movement of insects contributes to capture probability [13, 21].

Much of the data currently available for tephritid movement come from either mark-release-recapture (MRR) or flight mills bioassays [22, 23]. However, MRR studies are often complicated by issues of low recapture rates, high resource and time requirements, replication validity and other statistical issues, and monitoring flies that either move the furthest or have the potential to do so [24]. Additionally, it is not straightforward to relate the movement of laboratory-reared flies, which are often used in MMR, to that of wild flies [24]. Flight mills have been used primarily to provide insights into tephritid flight speed, distance, and duration, along with the effects of other physiological factors [25–27]. However, interpreting the ecological relevance of flight mill results can be difficult as these are conducted in the laboratory in the absent of environmental cues [28, 29].

Beyond MRR and flight mills, tracking flies in the field allows a more direct method for determining movement parameters. While studies using direct visual observations of flies in trees have been conducted [30, 31], a more common approach with insects is to use a tracking device. A number of different tracking devices have been employed to study insect movement including radio frequency identification (RFID), radio telemetry (RT), and harmonic radar (HR) [32]. Relatively few Dipteran spp. have been studied using tracking devices [32] as flies are generally small- to medium-sized insects and require small, light tags. To our knowledge, all previous dipteran tracking studies have utilized HR (Tachinidae [33], Sarcophagidae [33], Tephritidae [34–37], Glossinidae [38]). HR tags have the advantage of generally being much lighter than RT tags, which must contain a battery, while having a detection range of up to 500 m depending on the radar unit employed [32, 39]. In addition to being small, HR tags are less expensive than RT tags and the lack of a battery allows both long shelf-life and field-life. However, the HR detection range is typically shorter than that of RT, tags aren't uniquely identifiable, and they are highly sensitive to both tag orientation relative to the radar unit and influences from the terrain and vegetation. There are two components of HR, 1) a radar transceiver unit, which both emits a directional microwave signal and 'listens' for a reflected signal at twice the broadcast frequency, and 2) a diode tag that receives the original microwave signal and reemits a frequency-doubled signal [40]. HR units can be stationary ground-based [39] or mobile, which includes handheld units [33, 41]. Previous studies have tracked insects, including several fly species [14], using handheld HR units manufactured for avalanche rescue by the RECCO corporation [40, 42, 43].

Herein we report the HR tracking of wild male melon flies in field cages, a papaya field, and open parkland in Hilo, HI. HR tags used in this study were fabricated using superelastic nitinol wire which is both lightweight and flexible, allowing for unimpeded fly movement while retaining maximum antenna length to improve detection range. Unexpectedly, individual flies showed directionally biased flight movements in both cage and field experiments.

## Materials and methods

### Insects

Wild male *Z. cucurbitae* were captured from a papaya field in Keaau, HI (19.6276, -155.0406). Small pieces of dental wick treated with cuelure (4-(3-oxobutyl)phenyl acetate) were placed in plastic containers (8 cm x 12 cm, cylindrical) which were set on the ground between papaya trees for 10–15 mins. Captured flies were transferred to screen cages (40 cm x 40 cm x 40 cm) and held under controlled conditions at the USDA-ARS-PBARC facility in Hilo, HI. Holding conditions were 24–27˚C, 65–70% RH, and a photoperiod of 12:12 h (L:D). Flies were

provided with water, honey (spread on the cage screen), and a solid diet of three-parts sucrose, one-part protein yeast hydrolysate (Enzymatic, United States Biochemical Corporation, Cleveland, OH, USA), and 0.5-part torula yeast (Lake States Division, Rhinelander Paper Co., Rhinelander, WI, USA).

## Harmonic radar

Dipole harmonic radar tags were fabricated from a Schottky diode (RECCO AB, Lidingö, Sweden) and nitinol wire. Two 4 cm lengths of wire were attached to the diode with UV activated adhesive (Bondic, Niagara Falls, NY, USA) so that each wire touched one of the diode contacts while avoiding the opposite diode contact and the other wire. Electrical connections between the wires and the diode contacts were secured using conductive silver paint (GC Electronics, Rockford, IL, USA). Nitinol wire from two manufacturers were used to make tags. Initial tags, used in experiment 2, were made with 0.0254 mm diameter superelastic nitinol wire purchased from Kellogg's Research Labs (Salem, NH, USA). This wire had a slight curvature which resulted in an overall slight S-shape to the tags. Tags for all other experiments were fabricated with straight annealed 0.0254 mm diameter superelastic nitinol wire purchased from Fort Wayne Metals (Fort Wayne, IN, USA). Early attempts at tag fabrication utilized tungsten (0.071 mm diameter), copper, and platinum (Wollaston process, 0.00254 mm diameter, Leico Industries, Inc., Lyndhurst, NJ, USA) wires which were not found to be suitable.

Tags were attached to the fly thorax in a longitudinal orientation (Fig 1) using a UV activated adhesive (Bondic, Niagara Falls, NY, USA). This was accomplished by chilling wild male melon flies for 20 mins before being held by their legs between the fingers. Tags were then dipped in the adhesive and positioned on the thorax prior to being cured with light from a UV LED. Care was taken not to glue the wings or the head during tag attachment. Initial flight testing was also conducted with the tags attached to the underside of the abdomen in a lateral orientation but this attachment position was observed to impede normal flight behaviors.

Harmonic radar transceiver units (R9) were purchased from RECCO AB (Lidingö, Sweden). Locating flies with HR was accomplished either by searching an area to which a fly was visually noted to have flown to or by moving around either cages or fields in a regular pattern. During searching, the RECCO unit was rotated in the hand and moved from side to side to maximize signal detection by aligning the transceiver with the tag attached to the fly. Under optimal conditions, alignment of the RECCO unit with the tag, without vegetation interference, had a maximum detection range of approximately 10 m. For determining search patterns, a 5 m detection range was assumed.

## Flight ability testing (20 May 2021)

Experiment 1 tested the flight ability of tagged flies versus untagged flies. The experimental design was based on quality control measures outlined for tephritids by FAO/IAEA/USDA [44]. Twenty HR tagged and 20 untagged wild male melon flies were placed into chilled storage (4°C) for 5 mins in separate containers. Tags used in this experiment were fabricated from straight annealed superelastic nitinol wire with an approximate mass of 0.8 mg. Each fly was taken individually to the flight-testing room, which had a temperature of 26.5°C and 60% RH and dropped out of the container into the bottom of a PVC flight tube (20 cm x 14.5 cm x 15.5 cm) inside a cage (30.5 cm x 30.5 cm x 76 cm). The time for the fly to become responsive and exit the tube was recorded. Two flights were recorded for each fly and the times were averaged. Flies were considered a non-flier after 15 mins.

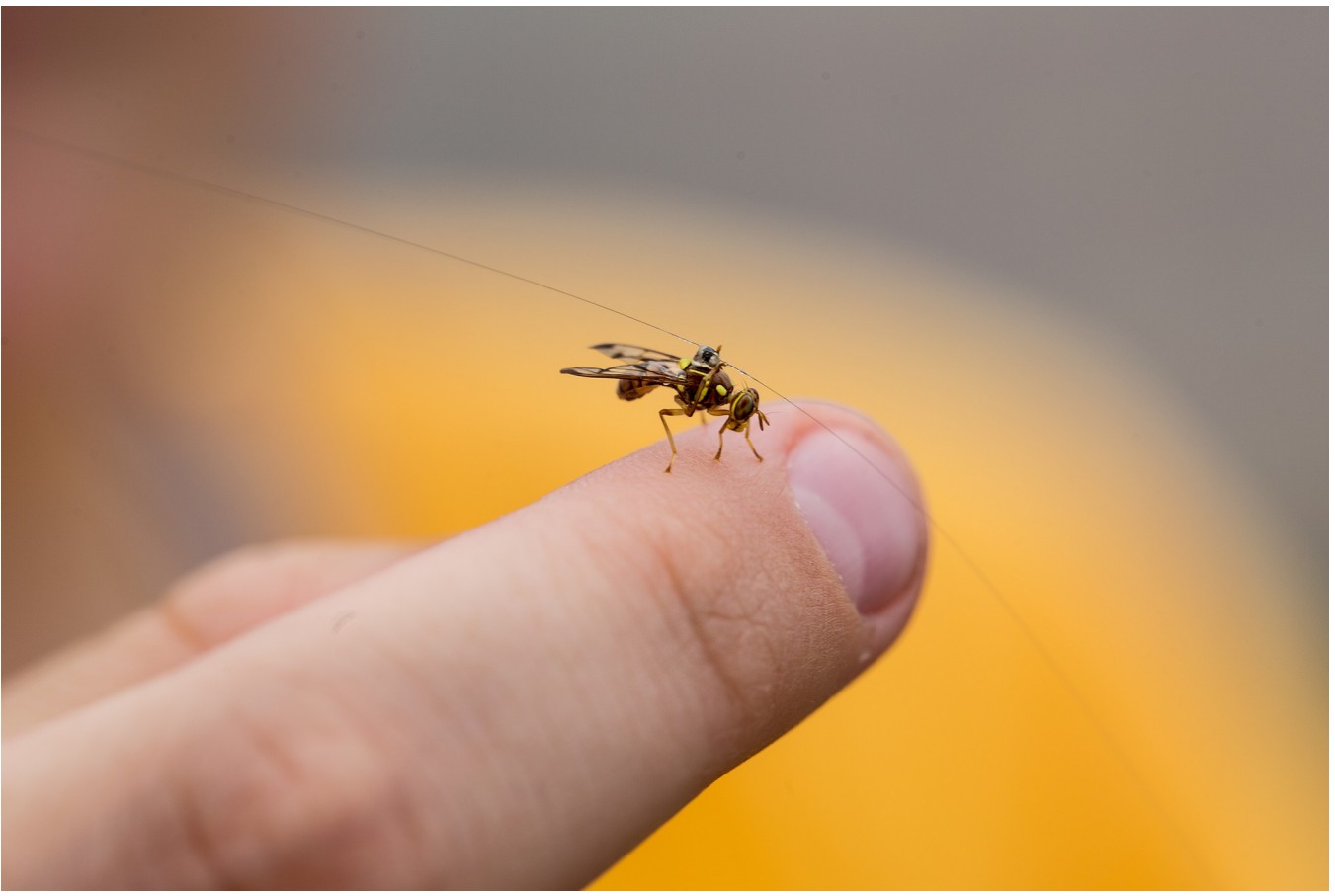

**Fig 1. A wild male melon fly, *Zeugodacus cucurbitae*, with a harmonic radar tag attached.** The tag shown weighed approximately 0.8 mg and was fabricated from a diode and superelastic nitinol wire.

### Field cage experiments

Experiment 2 (29 December 2020–5 January 2021) investigated the movement of tagged flies in outdoor field cages. In this initial experiment, tags made with non-annealed nitinol wire were used. Flies were released from the center of one of two outdoor field cages (15 m x 6 m x 2.5 m) located at USDA-ARS-PBARC in Hilo, HI. Ten flights were recorded for each fly. After each flight the landing position was recorded and the fly was recaptured and rereleased from the center of the cage. Step-distances (the length of a single flight from take-off to landing point), flight directionality (the angle from take-off to landing point), and pseudo-turning angles (the difference in angle between successive flight directions) were calculated from recorded fly positions. This initial experiment used slightly heavier tags (approximately 1.1 mg, only used in this experiment), wind information was not recorded, and three flies were tracked twice (N (tracking episodes) = 11).

Experiment 3 (12–25 May 2021) was conducted in a manner similar to experiment 2 except that slightly lighter tags (approximately 0.8 mg) made with annealed nitinol wire were used, wind direction and speed were recorded in the near vicinity of moving flies using a Kestrel 4500 Weather Meter (Kestrel Instruments, Boothwyn, PA, USA), and flies were run only once (N = 25).

## Papaya field experiment (28 May 2021–11 June 2021)

Experiment 4 investigated the movement of tagged flies in a papaya field in Keaau, HI (19.6280, -155.0407). The field was approximately 11,000 m$^2$ in size with papaya trees ranging in height from 2–6 m. The field was bordered on three sides by dense tropical forest. The fourth field border consisted of tall grasses and sparse trees. No permit was required for access to this field site. Flies were tracked one at a time after being released into the field. Papaya trees were used as release sites with new trees used for successive releases. No more than two tagged flies were tracked at the same time (separate crews followed each fly). After release, tagged flies were tracked through the field with landing locations (specific tree) recorded after each flight. At most landing locations the fly was visually located, however, sometimes the canopy height allowed the landing location to be identified only by signal detection. Flies were allowed to rest for 5 mins following each flight. If a fly had not flown again after 5 mins, it was gently poked or the surrounding foliage was disturbed to induce flight to another tree. Flies with at least five recorded flights were used in the analysis. Up to 10 steps were recorded for each tagged fly. When possible, flies were recaptured and removed from the field after the 10$^{th}$ recorded flight. Wind speeds and directions were recorded in the near vicinity of moving flies. Step-distances (flight distances), flight directionality (angle from take-off to landing), and turning angles (angle between successive flight directions) were calculated from recorded fly positions.

## Open field experiment (7 January 2021 and 8 June 2021)

Experiment 5 investigated the movement of tagged flies in an open field in a public park (Lokahi Park, approximately 25,000 m$^2$ in size, 19.6979, -155.0734) in Hilo, HI. No permit was required for access to this field site. Tagged flies were released from two release points (varying by date) at the center of an open grassy (mowed) field (approximately 10,000 m$^2$). The open field was bordered by a mixture of forest, tall grass, and basketball and tennis courts. Flies were tracked after release with landing locations recorded after each flight. When possible, flies were recaptured and released from the same central point. Wind speeds and directions were recorded in the near vicinity of moving flies. Step-distances and flight directionality were calculated from recorded fly positions.

## Statistical analysis

For experiment 1, the proportion of fliers to non-fliers was tested using Fisher's exact test. The time to exit the tube was compared by t-test calculated using Microsoft Excel (version 2108, Microsoft Corporation, Redmond WA).

For experiments 2–4, the Watson-Williams test for homogeneity of means was used to determine if the flight directions varied between flies. Subsequently the Rayleigh test and the Hermans–Rasson test [45] were used to determine if flight directions were random for each set of flights for an individual fly. Additionally, the V-test was used to assess the effect of wind direction on flight direction in experiment 3–5 [46]. All circular statistical analyses were performed using R packages CircStats, circular, and CircMLE [47]. Chi-squared for turning angles were performed in R [47]. For experiment 4, individual flights were grouped into quadrants and mean step distances were compared using ANOVA followed by Tukey's HSD ($\alpha = 0.05$) while the proportion of steps in each quadrant was analyzed using a contingency table approach followed by the Marascuilo procedure, both performed in R [47]. Equations for step frequencies vs. step-distances were calculated using Microsoft Excel.

## Results

For experiment 1, the flight ability experiment, the proportion of fliers (F) to non-fliers (NF) during the 15 mins period did not differ between tagged (38F:2NF) and untagged (40F:2NF) flies (Fisher's exact test, $P = 0.4937$). Likewise, the mean time to exit the flight tube did not differ between tagged (164 ± 29 sec) and untagged (129 ± 29 sec) flies (t-test, $P = 0.400566$).

Experiment 2, the initial field cage experiment, showed that when all flights were taken together, flight directions did not show directionality ($P = 0.9433$, Rayleigh test; $P = 0.2160$, Hermans-Rasson test) (S1 Fig). However, individual flies appeared to have strong directional preferences. Circular statistical analysis with the Watson-Williams test showed that flight angle means were not homogeneous ($F = 355.89$, df1 = 10, df2 = 99, $P < 0.001$). Further, the observed 10 flights for each individual fly were shown to have non-random directionality (Rayleigh test $P$-values ranged from 0.01 to < 0.001, Hermans-Rasson test $P$-values ranged from 0.019 to < 0.001, mean resultant lengths ($\bar{R}$) ranged from 0.691 to 0.936) (Fig 2). Combined pseudo-turning angles (Fig 3A) were non-random ($\bar{R} = 0.699$, $P < 0.001$, Rayleigh test, $P = 0.001$, Hermans-Rasson test), showed no right-left bias ($P = 0.837$, chi-squared test), but did show a pronounced bias towards moving within 90˚ left or right of the directly previous flight ($P < 0.001$, chi-squared test). The mean step-distance (distance from take-off point to landing point) was 3.6 ± 0.2 m with a median of 3.2 m (N = 110, Fig 4). Flies appeared to preferentially land on the sides or roof of the screen cage although some also landed in the mowed grass within the cage. Different flight courses were observed including direct flights from the take-off to landing to more circuitous flight paths in which the fly flew in circles or changed directions multiple times before landing.

In Experiment 3, when all flights were taken together, flight directions were not homogeneous but showed directionality ($P < 0.001$, Rayleigh test; $P < 0.001$, Hermans-Rasson test) (S1 Fig). Additionally, V-test for all flights show a unimodal distribution positively correlated with the mean wind direction (flies preferentially flew downwind) ($P < 0.001$) (S1 Fig). Again, as observed in experiment 2, the Watson-Williams test showed that flight angle means were not homogeneous ($F = 40.865$, df1 = 24, df2 = 225, $P < 0.001$). Similarly, 22 of the 25 flies showed non-random flight directionality by the Rayleigh test ($P$-values ranged from 0.489 to < 0.001) (Fig 5). Twenty-one of the 25 flies showed non-random flight directionality by the Hermans-Rasson test ($P$-values ranged from 0.737 to < 0.001) (Fig 5). Combined pseudo-turning angles (Fig 3B) were non-random ($\bar{R} = 0.619$, $P < 0.001$, Rayleigh test, $P = 0.001$, Hermans-Rasson test), showed no right-left bias ($P = 0.442$, chi-squared test), but did show a pronounced bias towards moving within 90˚ left or right of the directly previous flight ($P < 0.001$, chi-squared test). Three reps had no wind recorded, 9 flies showed flight directionality correlated with the wind direction, and 13 flies showed flight directionality uncorrelated with the wind direction by a V-test (Fig 5). The mean step-distance was 4.1 ± 0.1 m with a median of 3.5 m (N = 250) (Fig 4).

Experiment 4, the papaya field experiment, showed that when all flights were taken together, flight directions were not homogeneous but showed directionality ($P < 0.001$, Rayleigh test; $P < 0.001$, Hermans-Rasson test) (Fig 6). Further, flight directions show a unimodal distribution positively correlated ("downwind") with the mean wind direction ($P < 0.001$, V-test) (Fig 6). The number of steps per quadrant were unequal (contingency table approach, $\chi^2_{exp}$ (22.867) > $\chi^2_{crit}$ (16.266), df = 3, $P < 0.001$) with the quadrant opposite the wind (3rd quadrant of S2 Fig) having the highest number of steps (Marascuilo procedure). Mean step distances also differ by quadrant (ANOVA ln(x) transformed), $F = 8.434$, df = 3, $P < 0.001$) with the 3rd quadrant having the numerically highest value (ANOVA, Tukey's HSD, S2 Fig).

Again, as was observed in experiments 2 and 3, the Watson-Williams test showed that flight angle means were not homogeneous ($F = 12.571$, df1 = 19, df2 = 140, $P < 0.001$). Similarly, 12

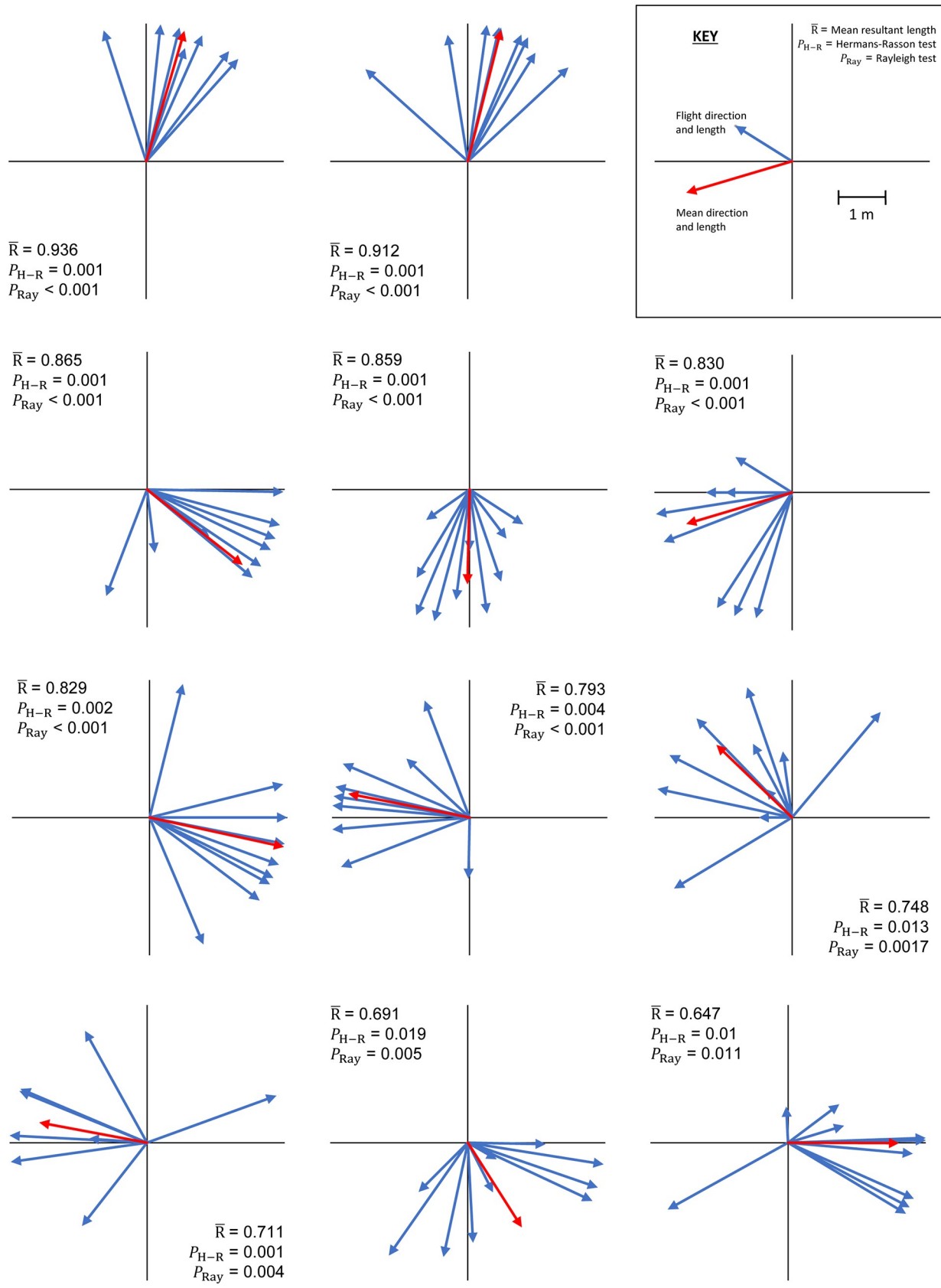

**Fig 2. Flight directions and lengths of HR tagged *Zeugodacus cucurbitae* for experiment 2 (field cage).** Each replicate consisted of a series of 10 flights with a single tagged fly. Blue arrows represent individual flights, while red arrows show the mean flight direction and length.

of the 20 flies showed non-random flight directionality by the Rayleigh test (*P*-values ranged from 0.539 to < 0.001) (Fig 7). Ten of the 20 flies showed non-random flight directionality by the Hermans-Rasson test (*P*-values ranged from 0.422 to 0.001) (Fig 7). Combined turning angles (Fig 8) were non-random ($\bar{R}$ = 0.419, *P* < 0.001, Rayleigh test, *P* = 0.001, Hermans-Rasson test), showed no right-left bias (*P* = 0.735, chi-squared test), but did show a pronounced "forward" movement bias (*P* < 0.001, chi-squared test). Ten flies showed flight directionality correlated with the wind direction while the other 10 did not (*P*-values ranged from 0.995 to < 0.001, V-test) (Fig 7). Overall, the mean step distance for experiment 4 was 6.0 ± 0.5 m with a median of 3.8 m (N = 160, Fig 4). When step-distances less than 1 m are removed, flight distances in papaya are well described by the power equation, step freq. = 0.291 x step dist.$^{-1.13}$ ($R^2$ = 0.727, S3 Fig). Step-distances were categorized into 1 m intervals for this analysis. Step-distance less than 1 m were likely undercounted as these flights generally kept the fly within the same tree and were therefore not recorded following the experimental protocol. Flies generally flew into the foliage of the papaya trees although some did land on tree trunks or on the ground. As tree height varied throughout the field so did the relative landing height of the flies.

Experiment 5, the open field experiment, showed that when all flights were taken together by date, flies released on 7 January showed a non-random flight directionality by the Rayleigh test (*P* = 0.0397) but not by the Hermans-Rasson test (*P* = 0.059). Flies released on 8 June showed a non-random flight directionality by both the Rayleigh test (*P* = 0.0248) and by the Hermans-Rasson test (*P* = 0.002). Further, flight directions show a unimodal distribution positively correlated with the mean wind direction on both 7 January (*P* = 0.008, V-test) and 8 June (*P* = 0.003, V-test) (Fig 9). The mean step-distance was 6.0 ± 0.5 m with a median of 3.8 m (N = 160) (Fig 3). Flight distances in the open field experiment showed no correlation

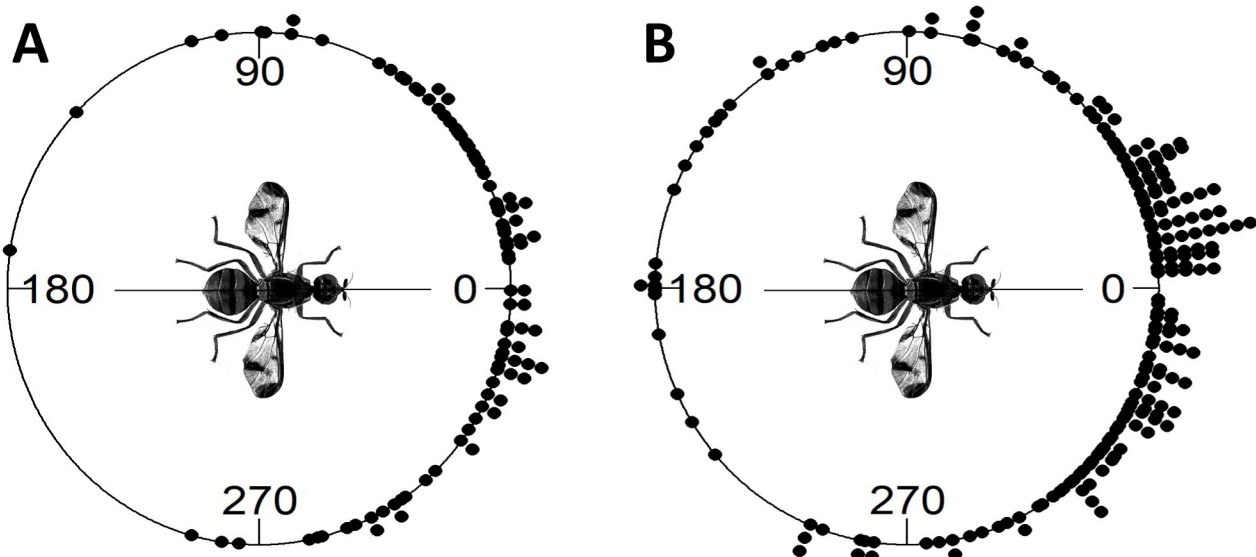

**Fig 3.** Combined pseudo-turning angles of HR tagged *Zeugodacus cucurbitae* for experiment 2 (A) and experiment 3 (B). A turning angle of zero indicated that a fly flew in the same direction as the directly previous flight. Combined turning angles were non-random by both Rayleigh and Hermans-Rasson tests, showed no right-left bias, but indicate a pronounced bias towards moving within 90° left or right of the directly previous flight.

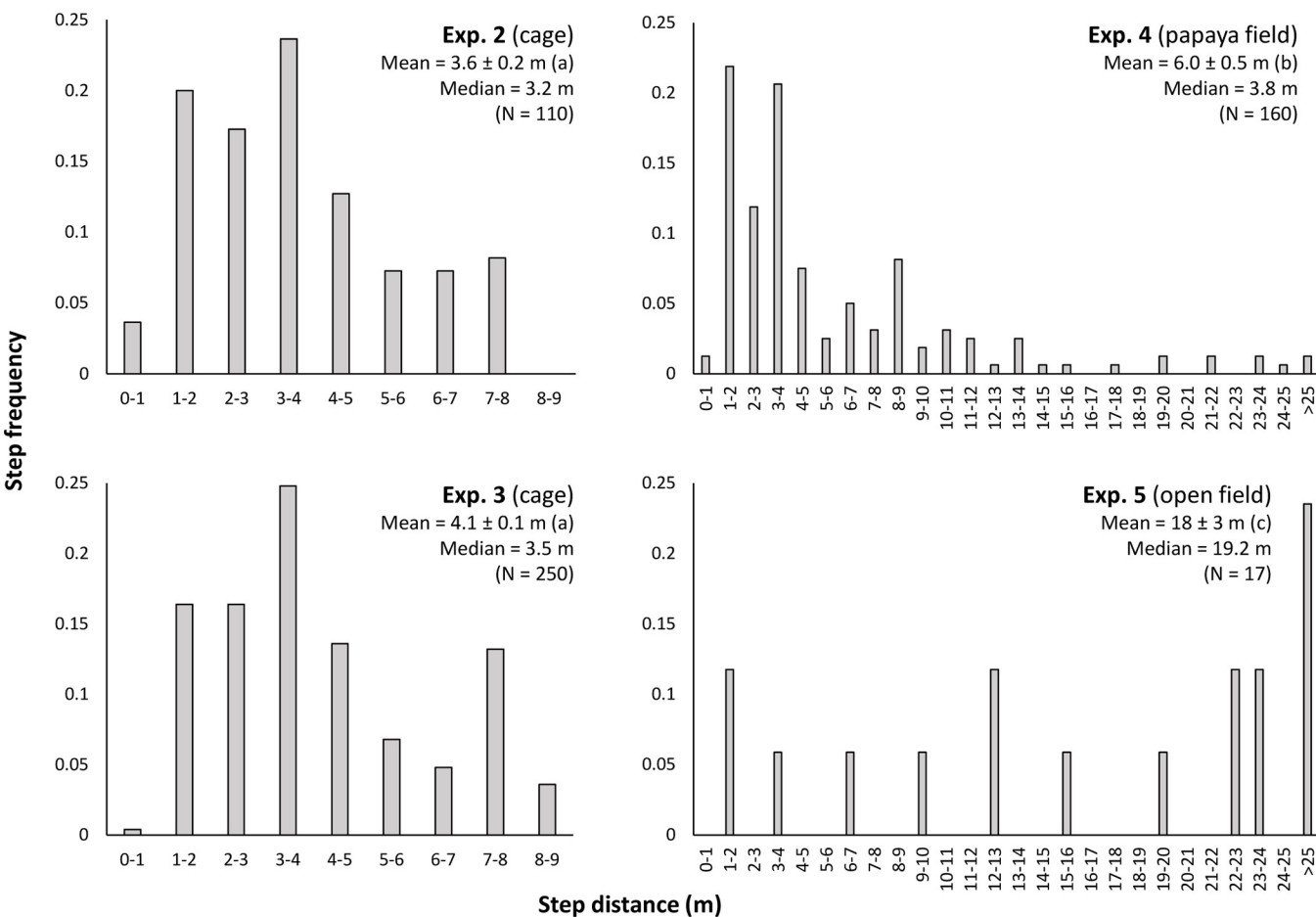

**Fig 4. Melon fly, *Zeugodacus cucurbitae*, flight step-distances for experiment 2–5.** Different letters, in parentheses, indicate significant differences by ANOVA followed by means separations with Student-Newman-Keuls ($P$ = 0.05).

between step frequency and step distance (S3 Fig). Tagged flies were generally observed to alight on the first tall grass or tree foliage they encountered (Fig 9). The exception to this observation was flies whose flights paths were not long enough to encounter tall grass of trees and these flies landed on the mowed grass.

## Discussion

### Harmonic radar tags

Tag mass and size [48], tag entanglement [49], and antenna deformation [48] have all presented obstacles to HR tracking of insects. HR tags are required to be extremely small and light for this application, with tag mass less than 5% of the insect mass, although the empirical basis for this limit is weak [32]. Additionally, tag antennas need to be flexible enough so as not to impede movement while simultaneously avoiding permanent deformation when bent. Copper [42], copper-cladded steel [41], silver-plated copper [50], steel [50], and aluminum [33, 43] wires have been used as antennas for HR tags in previous studies with occasional reports of wire deformation [41, 51] and entanglement and/or encumbrance [43, 50].

The successful tracking of melon flies in this study was largely due to the use of superelastic nitinol wire as the antennas for the HR tags. Nitinol is light, relatively inexpensive, and

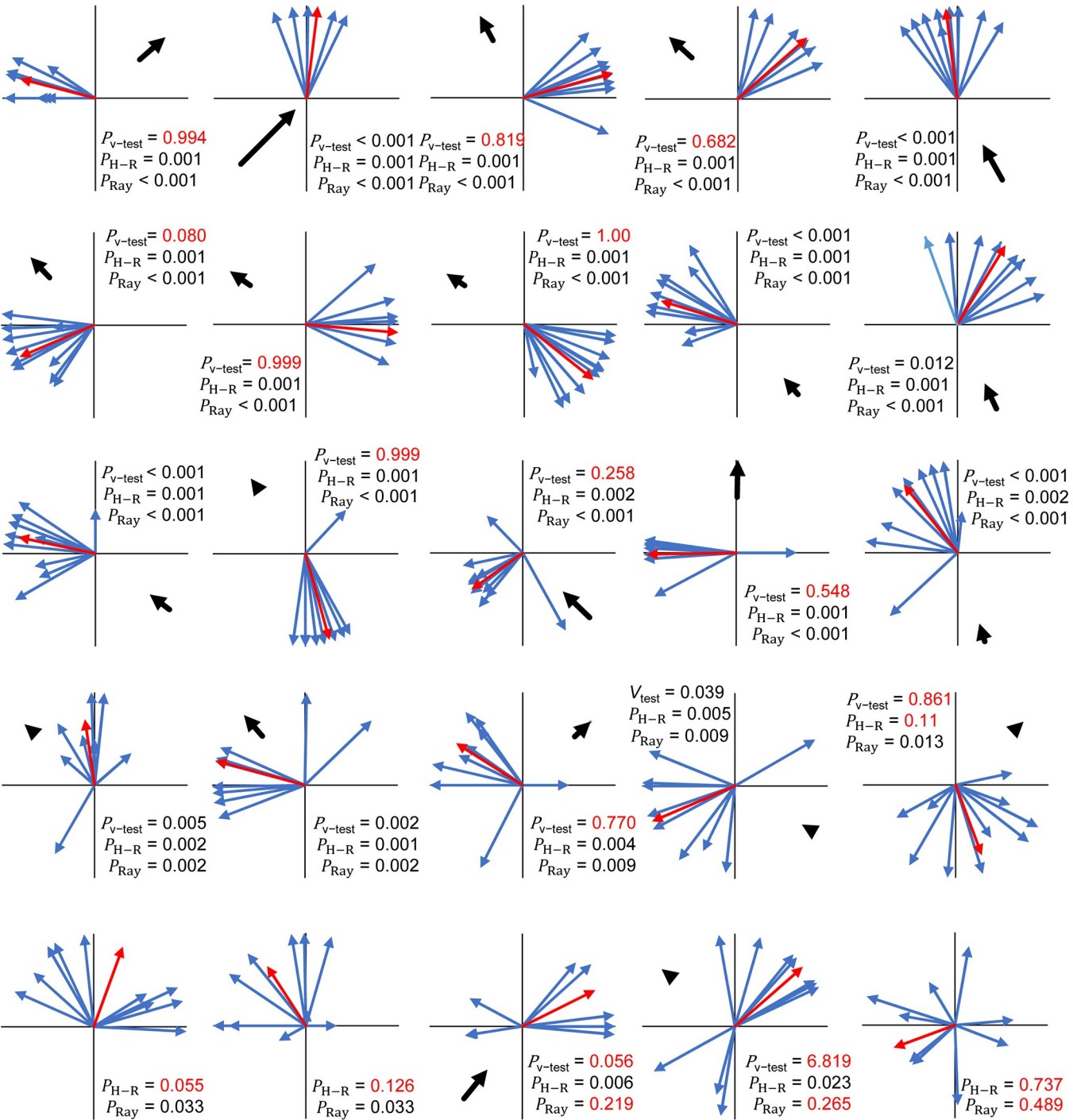

**Fig 5. Flight directions and lengths of HR tagged *Zeugodacus cucurbitae* for experiment 3 (field cage).** Each replicate consisted of a series of 10 flights with a single tagged fly. Blue arrows represent individual flights, while red arrows show the mean flight direction and length.

possesses the conductive properties needed to operate as an HR antenna. The superelasticity of nitinol allows this nickel-titanium alloy to undergo substantial deformations (up to 30 times greater than other metals) before returning to its original undeformed shape [52]. The advantages of nitinol were clearly seen when compared with the other two metal wires used in this

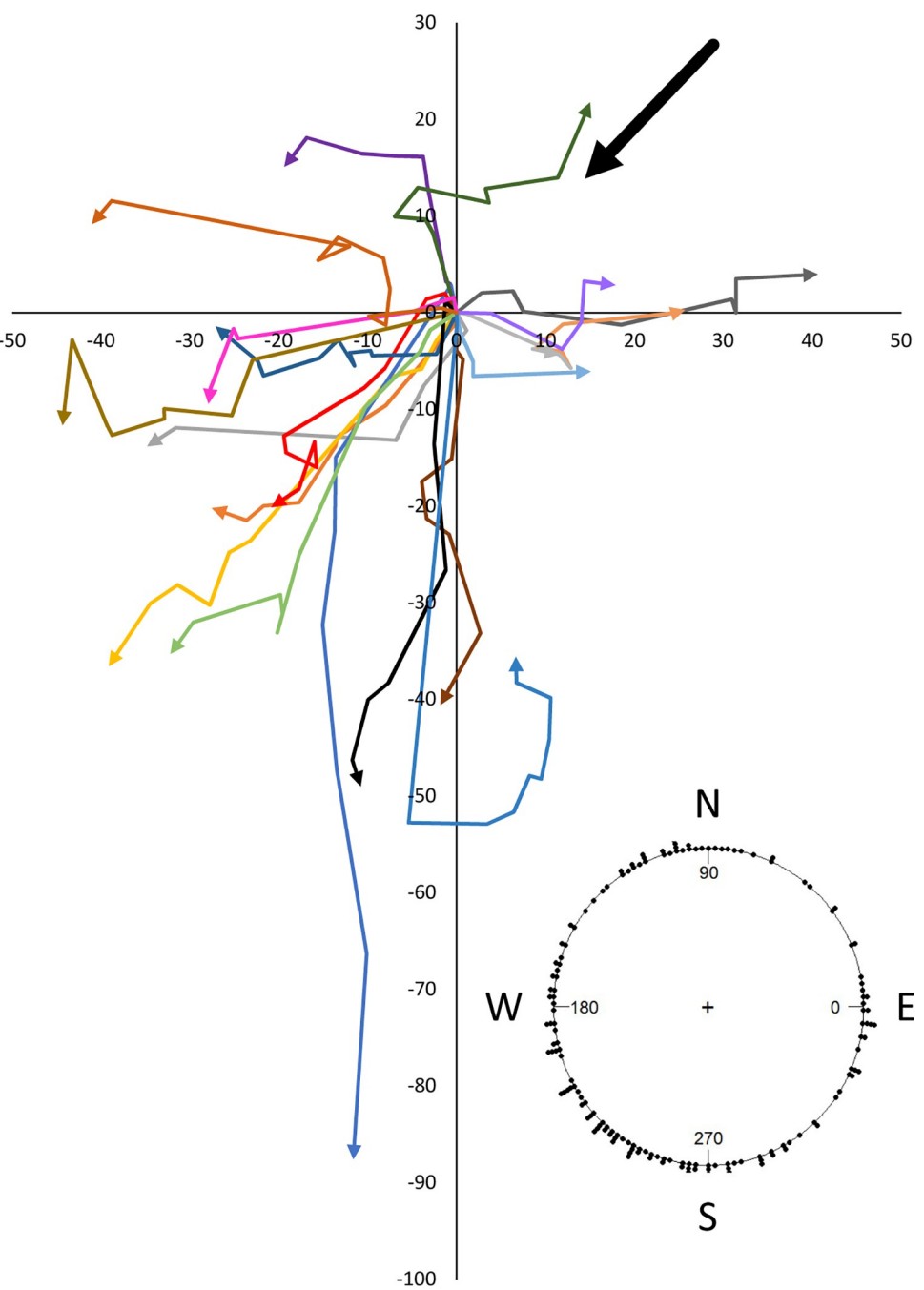

**Fig 6. HR tagged *Zeugodacus cucurbitae* flight directions and lengths for experiment 4 (papaya field).** Colored arrows represent a series of 5–10 flights for a single tagged fly. The large black arrow shows the mean wind direction for the duration of tracking. When all flights were taken together, flight directions were not homogeneous but showed directionality ($P < 0.001$, Rayleigh test; $P < 0.001$, Hermans-Rasson test) and a unimodal distribution correlated with the mean wind direction ($P < 0.001$, V-test).

study. Wollaston process wire, a platinum wire clad in silver, was light and flexible but would quickly become deformed by bending and tangle, thus critically reducing the detection distance of the tags. In contrast, Tungsten wire is one of the hardest and most dense metals. While antennas made with tungsten did not deform or tangle, they restricted the movement of

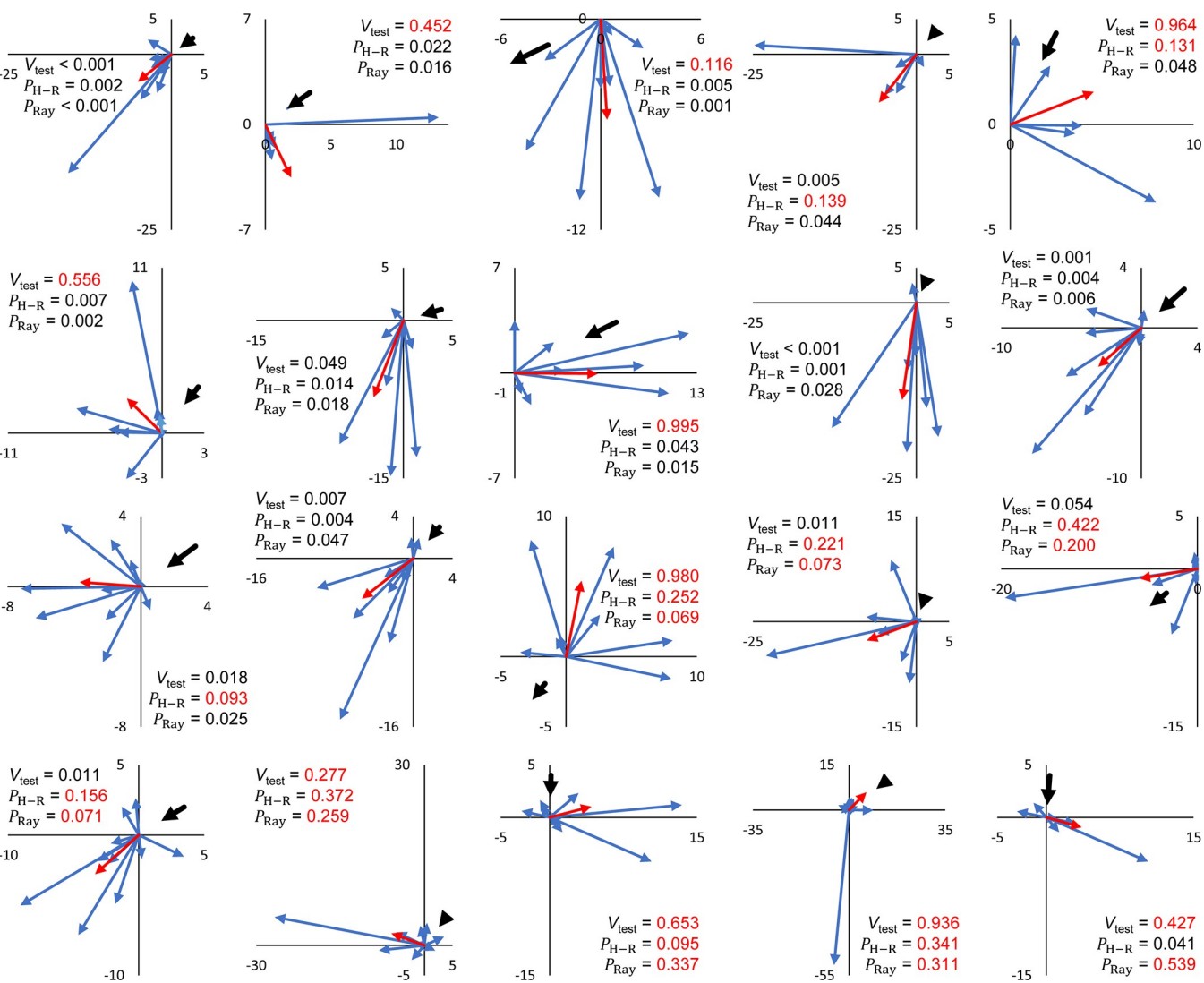

**Fig 7. Flight directions and lengths of HR tagged *Zeugodacus cucurbitae* for experiment 4 (papaya field).** Each replicate consisted of a series of 5–10 flights with a single tagged fly. Blue arrows represent individual flights, while red arrows show the mean flight direction and length.

the flies due to the inflexibility of the forward-facing wire, which was observed to obstruct forward movements of the fly. Also, given the density of tungsten, tags made with this wire were heavier than both those made with nitinol or Wollaston process wire.

There are a limited number of previous studies using HR to track dipteran species [32], likely due to their small size and lack of adaptation to carrying loads in flight. Roland et al. [33] were the first to demonstrate that it was possible to track flies using the tachinid fly *Patelloa pachypyga* (Aldrich & Webber) and the sarcophagid *Arachnidomyia aldrichi* (Parker).

Larger Coleoptera (many of which were tracked as they walked) and Hymenoptera have been more commonly tagged [32]. Interestingly, the other dipteran species that has been studied in the field with any depth is a tephritid fruit fly, the Chinese citrus fruit fly, *B. minax* (Enderlein) [35]. This species is among the largest tephritids (approximate male weight 44 mg) and is roughly three times the mass of the melon fly (approximate male weight 15 mg). The work with *B. minax* utilized a HR tag weighing 3.4 mg, which is more than four times the mass

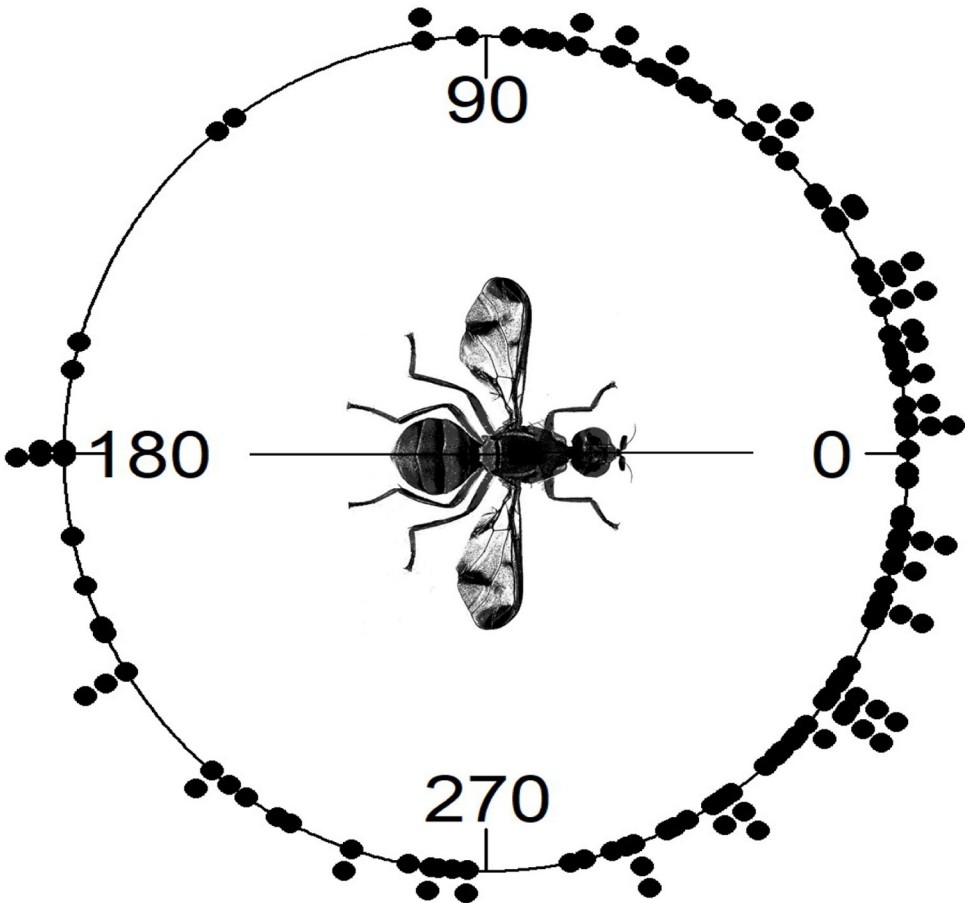

**Fig 8. Combined turning angles of HR tagged *Zeugodacus cucurbitae* for experiment 4 (papaya).** A turning angle of zero indicated that a fly flew in the same direction as the directly previous flight. Combined turning angles were non-random ($\bar{R}$ = 0.419, $P$ < 0.001, Rayleigh test, $P$ = 0.001, Hermans-Rasson test), showed no right-left bias ($P$ = 0.735, chi-squared test), but showed a pronounced "forward" movement bias ($P$ < 0.001, chi-squared test).

of the tags in the current study (0.8 mg). However, the tag to fly mass ratios are similar in both studies with each being around 5% of the total insect body mass. Tags used on *B. minax* were roughly 8% of the fly body mass with repeated experiments finding little effect on the flight characteristics of the fly [34, 36, 37]. Although the flight testing in the current study was not as extensive as that carried out with *B. minax*, we similarly found minimal negative effect on *Z. cucurbitae* flight (experiment 1).

## Flight directionality

Melon flies tracked in this study showed both individual level flight directional biases (Figs 2, 5, and 7) and, in some experiments, collective directional biases, both turning angles (Figs 3 and 8) and combined absolute flight directions (Figs 6 and S1). At the individual level in field cage experiments (experiments 2–3), 30 of the 36 flies observed (83%) showed directionally biased flights while similar biases were observed in roughly half the flies tracked in a papaya field (experiment 4). Additionally, mean flight directions varied between flies (Watson-Williams test) in experiments 2–4 showing strong inter-individual differences in directional orientation. These individual-level flight directional biases may be an example of biased behaviors

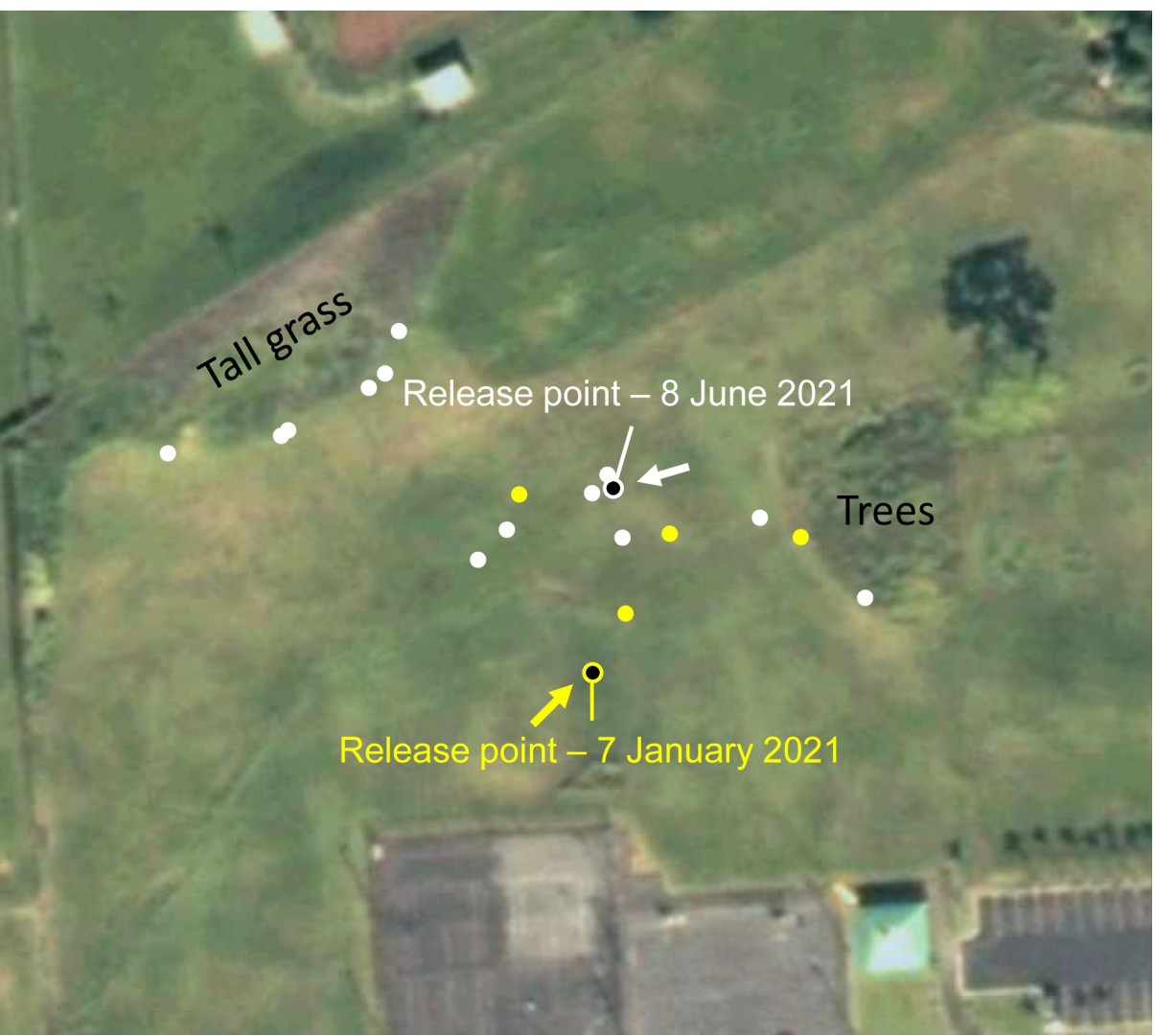

**Fig 9. Tagged melon fly, *Zeugodacus cucurbitae*, release points (black center with yellow or white border) and landing points (yellow or white dots) at Lokahi Park, Hilo, HI.** Yellow and white arrows indicate the mean wind direction during fly releases. Source: U.S. Geological Survey, 2011, USGS High Resolution State Orthoimagery for the East Side of Hawaii Island: 05QKB820780_200912_0x5000m_CL_1: U.S. Geological Survey.

in insects, such as turning biases, leg preferences, and antennal asymmetries. These are linked to lateralization in the central nervous system [53, 54]. Insect individual-level bias include turning bias in the human body louse [55], a staphylinid [56], bumblebees [57], 7-spot ladybird beetles [58, 59], drosophila [60], and honeybees [61], as well as forelimb lateralization in desert locusts [62] and the biased use of laterally paired penises in earwigs [63].

There are examples of insect population-level turning biases which include giant water bugs [64], several ant spp. [65–67], cockroaches [68], honeybees [69], and bumblebees [70]. Population-level biases have also been found in a number of diptera including turning in drosophila [60], and mating behaviors in *Culex pipiens* L. [71] as well as two tephritids, *B. oleae* (Rossi) [72] and *C. capitata* [73]. These left-right asymmetries, either at the individual or population level, demonstrate innate behavioral biases which in many instances relate to locomotion and more specifically turning. However, while these left-right movement associated biases share may similarities with the movement biases observed with male melon flies in this study,

the complexity of 360˚ flight directional orientation is likely higher than a simpler binary left-right choice.

Similar observations of individual insects displaying biased directional flight can be seen in the migratory butterfly *Pieris brassicae* L. Individual butterflies of subsequent generations have been shown to fly in opposite directions, the spring generation generally flies north while the autumn generation flies south, both in field observation [74–76] and cage bioassays [77–81]. Larranaga et al. [81] found that *P. brassicae* butterflies show both between and within individual variations in flight directional biases strikingly similar to those observed in the field cage experiments in the present study. Larranaga et al. [81] propose that inter-individual flight direction variation indicates that directional orientation is a heritable trait and that the migratory habit and directional orientation may be rapidly lost without selection, as was seen for butterflies bred for multiple generations in captivity.

While the *P. brassicae* cage experiment results [81] are similar to those observed in this study melon flies are not migratory and it is unclear what mechanism might be driving the observed directional flights. Perhaps the behavior leads to a greater dispersal of the entire fly population as flies having different directional biases would be more likely to move away from each other, minimizing competition.

Strongly biased directional melon fly movement can also be seen in the observed turning angles (Fig 8) and pseudo-turning angles (Fig 3). The term 'pseudo-turning angle' is adopted here to highlight that these flies were physically moved (returned to the release point) between successive flights. The observation that flies in the field cages successively choose similar flight directions is even more striking when considering that flies faced a random direction when returned to the release point (no attempt was made to orient them in the same direction as they were captured on landing). Flies making successive flights (steps) in the papaya field might show a "forward bias" simply because they continue is the same direction after landing (Fig 8). Flies in the field cage experiments had to reorient after being replaced at the release point and yet, as a group, flies chose subsequent flight directions within 90˚ left or right of the directly previous flight (Fig 3). Turning angle biases observed throughout this study may be examples what is known as 'persistence' or 'forward persistence' [82], the tendency observed in many animals towards forward movement [83–85]. Correlations between successive step orientations led to the development of the correlated random walk model (CRW) [82] and successive models such as biased random walk (BRW, consistent bias in a preferred direction or towards a target), and biased and correlated random walk (BCRW) [85]. These models have been applied to tephritid movement previously [30, 86] and the biased turning angle data observed for melon fly in this study support the use of modeling techniques that incorporate persistence and/or bias when modeling tephritid movement in the future.

Another possibility is that the observed flight directional biases are the result of flights being in an 'aroused state' from being handled and that the ballistic flight paths taken by flies are a mechanism to escape a dangerous situation. This hypothesis is supported by the fact that flies were subjected to handling or disturbances to induce movement throughout the experiment in which directionally biased movement was observed. In this case, future tracking studies that are less agitating to the flies should show greater randomness in the directions of flights, thus flight paths should look less like escape movement.

Collectively, when all flight (cardinal) directions are combined and analyzed for each of the experiments 2–5 (Figs 5 and S1). Flight directions are random for experiment 2 (first cage experiment), suggesting that flies, as a group, are not orienting towards one strong directional cue (e.g. visual, light, wind cues), but they do show directionality for experiment 3 (second cage experiment), experiment 4 (papaya field), and experiment 5 (open field) so directional cues may be at play in these experiments. Interestingly, a similar collective biased directional

dispersal behavior was observed for *B. minax* [35] with the biased *B. minax* movement attributed to flies moving out of an orchard and into an adjoining forest. The effect of environmental cues on insect movement are notoriously difficult to document and analyze [87]. Visual cues in the cage experiments were likely less intense and varied when compared to the papaya and open field experiments. The cage walls were made of white screen which obscured the view outside the cage. The cages were also erected in a gravel parking lot which further reduced the complexity of short-range cues. Light cues were largely similar for experiments 2–5 as all were conducted from late-morning through mid-afternoon. Variations in shade were greatest in the papaya field experiment.

No attempt was made to quantify either visual or light cues during this study. In contrast, wind speed and direction were recorded for experiments 3–5. Wind speeds and directions were recorded in the near vicinity of moving flies but may not reflect the actual microconditions experienced by the flies. Additionally, mean wind values were used for V-tests and while the variation in wind speed and direction were relatively low, this likely further weakens the ability to correlate wind parameters with fly movements. Even with these caveats, the mean wind direction was correlated with mean flight directions for both some individual flies and for combined flights from experiments 3–5. Where there was a correlation downwind movement was generally observed (negative anemotaxis). The observed mix of relationships between flight directions and wind, both flies with mean flight directions that correlate with the wind direction (19/42) and those that don't correlate (23/42), suggest that wind direction is one factor influencing fly flight direction but not the only factor.

Unsurprisingly, wind affects flight directionality for many tephritids but the magnitude of this effect and the interaction with other contributing factors is still largely unknown. Mean flight direction has been found to correlate with mean wind direction in studies of olive fruit fly, *B. oleae* [88], medfly [89], and Mexican fruit fly, *Anastrepha ludens* (Loew) [89]. However, a study conducted by Plant and Cunningham [90] observed the opposite–that medfly dispersal and wind were not related. Additional studies with Queensland fruit flies, *B. tryoni* (Froggatt) and *A. ludens* suggest that the relationship between flight direction and wind is complicated by environmental factors [91, 92] and wind speed [93].

## Flight distances

Step-distances (meters per flight) for male melon flies differed between the field cage, papaya field, and open field experiments with the longest mean flight distances observed in the open field (Fig 3). Step-distances were truncated in the two field cage experiments as flies were limited to maximum flight lengths of 3–9 m dependent of the direction of flight (flies would encounter the screen wall at these distances). The most common step-distances in experiment 2 and 3 (field cages) fell between 3–4 m and the mean step-distances were 3.6 ± 0.2 m and 4.1 ± 0.1 m respectively, which largely reflects instances of flies landing on the nearest cage wall. Given the limited possible maximum flight distances in the cages, it is difficult to draw conclusions about mean step-distances from these experiments.

In contrast, step-distances recorded in experiment 4 (papaya field) are not truncated. The most common step-distances in the papaya field were from 1–2 m with a mean step distance of 6.0 ± 0.5 m. The relationship between step frequency vs step-distance (categorized into 1 m intervals) was well described by a power function (S3 Fig) showing that flies generally make short flights between nearby trees with occasional longer flights. This result differed from what was seen in experiment 5 (open field) where there was no relationship between step frequency and step-distance (S3 Fig). Step-distances in the open field had the highest mean, 18 ± 3 m, with flights ending either in the first tall grass or trees encountered or on the ground if the

flight did not reach dense foliage. Flies in the open field were harder to track given the longer flights and tagged flies were lost at a higher rate when compared with experiments 2–4. Similar higher dispersal rates of melon flies in less favorable environments (fewer roosting sites and host plants) have been previously noted [94].

For *B. minax* the mean step-distances ranged from 2.3 ± 0.4 m to 6 ± 5 m depending on the experiment [35]. This is similar to the 6.0 ± 0.5 m step-distance found for melon fly in a papaya field (experiment 4). In the current study, flies were disrupted to keep them moving, which differs from the *B. minax* experiments and likely accounts for the longer total distance observed with melon fly. *Bactrocera minax* were never observed to move more than 25 m from the release point while this was routinely observed in melon fly (Fig 6).

While melon flies have been recorded to move up to 200 km [95], when host plants are plentiful, overall fly movements are thought to be nondispersive [24, 96, 97]. Melon flies are generally thought to show two types of movement, long-distance dispersal movement after emergence and more localized daily movement patterns in and around crop fields [96, 97]. Given that many detection and control efforts are based on understanding how melon fly, and other pest tephritids, move within the landscape, numerous previous studies have sought to quantify fly dispersal parameters [24, 94, 95, 98–101]. Additional studies have reported similar movement behavior for other pest tephritids such as *B. dorsalis* [99, 102–104], *B. tryoni* [93, 105–107], *B. latifrons* (Hendel) [108], *B. oleae* [88], *C. capitata* [89, 90], *Rhagoletis mendax* Curran [20, 86], and *Anastrepha* spp. [89, 92, 109]. While flight direction and distance were assessed in the current study, the timing to fly movements were artificially accelerated (disturbance to induce flight) making direct comparisons of flight parameters with previous MMR results complicated. Future tracking studies in which tagged flies are allowed to move more naturally (without artificial disturbance) are needed.

The data obtained in this study will be very valuable to melon fly detection and control. This is primarily because they enable application of more realistic and accurate models of melon fly movement such as the CRW [110, 111]. This model includes rare events and is an individual-level model of movement, compared with the more commonly used diffusion models which produce the distributional outcome of the insect's movement process but don't model the movement itself [112, 113]. Delimitation arrays can be optimized for improved response following an incursion, and quarantine and eradication boundaries can be set more effectively. In the event a melon fly is caught in a surveillance trap, the potential origin of that fly can now be better set to a given radius, enhancing the likelihood of an effective response.

## Conclusions

This study demonstrates the feasibility of tracking individual melon flies, a highly mobile, medium-sized flying insect, in both field cages and in the wild. The development of small, light-weight tags with flexible antennas was critical to successfully tracking of the flies with no detectable impact on flight ability. We observed an unexpected individual bias in flight directionality during repeated observations of fly movements. Along with flight directions, we were also able to record step-distance in field cages and in the wild. These observations suggest that the environment, papaya field vs. open parkland, substantially impacts the mean flight distance of tracked flies. Turning angle and step-distance are the primary parameters used in many insect movement models and the values reported for these parameters in this study may help to refine modeling for pest outbreaks.

## Supporting information

**S1 Fig.** Combined fly flight directions for experiments 2 (A) and 3 (B). Black arrow: wind direction; +: Release point. Flight directions were homogeneous for experiment 2 (*P* = 0.9433,

Rayleigh test; $P = 0.2160$, Hermans-Rasson test) while experiment 3 flight directions were not homogeneous but showed directionality ($P < 0.001$, Rayleigh test; $P < 0.001$, Hermans-Rasson test). A V-test for experiment 3 showed a unimodal distribution correlated with the mean wind direction ($P < 0.001$).
(TIF)

**S2 Fig. Comparisons between number of steps and mean step distances in each quadrant (as shown in Fig 5).** For steps in quadrant 1, flies flew against the wind. For steps in quadrant 3, flies flew with the wind. For steps in quadrants 2 and 4, flies flew across the wind. Using a contingency table approach, the number of steps per quadrant were found to be unequal ($\chi^2_{exp}$ (22.867) $> \chi^2_{crit}$ (16.266), df = 3, $P < 0.001$). Quadrants marked by different Greek letter have significantly different proportions of steps (Marascuillo procedure). Mean step distances with different Arabic letters are significantly different between quadrants (ANOVA, Tukey's HSD).
(TIF)

**S3 Fig. Melon fly, *Zeugodacus cucurbitae*, flight step-distances for experiments 4–5 with best fit lines.** For experiment 4, step-distances of less than 1 m were removed.
(TIF)

**S1 Data.**
(XLSX)

## Acknowledgments

The authors would like to thank Dr. James M. Yoder for his assistance with GIS images. Mention of trade names or commercial products in this publication is solely for the purpose of providing specific information and does not imply recommendation or endorsement by the U.S. Department of Agriculture. The authors declare no competing interest. USDA is an equal opportunity provider and employer.

## Author Contributions

**Conceptualization:** Nicholas C. Manoukis, Matthew S. Siderhurst.

**Data curation:** Matthew S. Siderhurst.

**Formal analysis:** Matthew S. Siderhurst.

**Funding acquisition:** Nicholas C. Manoukis, Matthew S. Siderhurst.

**Investigation:** Nicole D. Miller, Theodore J. Yoder, Nicholas C. Manoukis, Lori A. F. N. Carvalho, Matthew S. Siderhurst.

**Methodology:** Nicole D. Miller, Theodore J. Yoder, Nicholas C. Manoukis, Lori A. F. N. Carvalho, Matthew S. Siderhurst.

**Project administration:** Nicholas C. Manoukis, Matthew S. Siderhurst.

**Resources:** Nicholas C. Manoukis, Matthew S. Siderhurst.

**Software:** Nicholas C. Manoukis, Matthew S. Siderhurst.

**Supervision:** Nicholas C. Manoukis, Matthew S. Siderhurst.

**Validation:** Matthew S. Siderhurst.

**Visualization:** Matthew S. Siderhurst.

**Writing – original draft:** Nicole D. Miller, Theodore J. Yoder, Nicholas C. Manoukis, Matthew S. Siderhurst.

**Writing – review & editing:** Nicole D. Miller, Theodore J. Yoder, Nicholas C. Manoukis, Matthew S. Siderhurst.

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
