## [Decision Letter · Decision Letter 0]

19 Sep 2022

PONE-D-22-21998Harmonic radar tracking of individual melon flies, Zeugodacus cucurbitae, in Hawaii: Determining movement parameters in cage and field settingsPLOS ONE

Dear Dr. Siderhurst,

Thank you for submitting your manuscript to PLOS ONE. After careful consideration, we feel that it has merit but does not fully meet PLOS ONE’s publication criteria as it currently stands. Therefore, we invite you to submit a revised version of the manuscript that addresses the points raised during the review process.

We look forward to receiving your revised manuscript.

Kind regards,

Ramzi Mansour

Academic Editor

PLOS ONE

Journal Requirements:

"This study was supported in part by ARS project 2040-22430-027-00D and a Kauffman and Miller Research Award, Daniel B. Suter Endowment in Biology, Eastern Mennonite University."

 "NCM, MSS, LAFN - This study was supported in part by ARS project 2040-22430-027-00D, United States Department of Agriculture, Agricultural Research Service, https://www.ars.usda.gov/

NDM, TJY - This study was supported in part by a Kauffman and Miller Research Award, Daniel B. Suter Endowment in Biology, Eastern Mennonite University, https://emu.edu/science-seminars/daniel-b-suter-endowment

5. We note that Figure 9 in your submission contain satellite image which may be copyrighted. All PLOS content is published under the Creative Commons Attribution License (CC BY 4.0), which means that the manuscript, images, and Supporting Information files will be freely available online, and any third party is permitted to access, download, copy, distribute, and use these materials in any way, even commercially, with proper attribution. For these reasons, we cannot publish previously copyrighted maps or satellite images created using proprietary data, such as Google software (Google Maps, Street View, and Earth). For more information, see our copyright guidelines: http://journals.plos.org/plosone/s/licenses-and-copyright.

a) You may seek permission from the original copyright holder of Figure 9 to publish the content specifically under the CC BY 4.0 license.  

Reviewers' comments:

Reviewer's Responses to Questions

**Comments to the Author**

1. Is the manuscript technically sound, and do the data support the conclusions?

Reviewer #1: Yes

2. Has the statistical analysis been performed appropriately and rigorously? 

Reviewer #1: Yes

3. Have the authors made all data underlying the findings in their manuscript fully available?

Reviewer #1: Yes

4. Is the manuscript presented in an intelligible fashion and written in standard English?

Reviewer #1: Yes

5. Review Comments to the Author

Reviewer #1: Interesting information that warrants publication. I strongly suggest to reduce the Discussion, it is much too long. Firstly, please discuss only results relevant to male flies, you did not release female flies as far as I could tell. Therefore, remove any references to oviposition etc... Secondly, please reduce the discussion by removing unnecessarily lengthy comparisons with other literature. For example, discussion on wind direction: State that the following references found flies did orientate with wind direction, while the following did not. You do not need to explain what each reference found in detail. I would also give the whole document another detailed read to remove spelling and grammatical errors, I am not sure I caught them all.

6. PLOS authors have the option to publish the peer review history of their article (what does this mean?). If published, this will include your full peer review and any attached files.

Reviewer #1: No

---

## [Author Response · Author response to Decision Letter 0]

24 Sep 2022

Responses to specific comments:

Associate Editor 

- We have re-checked and believe we are meeting all of PLOS ONE's style requirements. 

- The following text has been added to both the descriptions of the papaya and open field experiments, “No permit was required for access to this field site.”

- We have resolved the mismatch.

4. Thank you for stating the following in the Acknowledgments Section of your manuscript…

- Funding information has been removed from the Acknowledgement. We would like our Funding Statement to read as follows,

“"NCM, MSS, LAFN - This study was supported in part by ARS project 2040-22430-027-00D, United States Department of Agriculture, Agricultural Research Service, www.ars.usda.gov/

NDM, TJY - This study was supported in part by a Kauffman and Miller Research Award, Daniel B. Suter Endowment in Biology, Eastern Mennonite University, emu.edu/science-seminars/daniel-b-suter-endowment

There was no additional external funding received for this study.

There is no award number for the Kauffman and Miller Research Award.

5. We note that Figure 9 in your submission contain satellite image which may be copyrighted…

- The Figure 9 has been replaced with a map that we independently generated that does not include any copyrighted material. The following text has been added to the Figure 9 legend, “Source: U.S. Geological Survey, 2011, USGS High Resolution State Orthoimagery for the East Side of Hawaii Island: 05QKB820780_200912_0x5000m_CL_1: U.S. Geological Survey.”

Reviewer 1 

General comments 

Firstly, please discuss only results relevant to male flies, you did not release female flies as far as I could tell. Therefore, remove any references to oviposition etc...

- The sentence in the discussion that mentions oviposition has been rewritten. We have examined the discussion to ensure that we do not discuss any female specific behaviors as the reviewer has rightly pointed out that we did not track females in this study.

I strongly suggest to reduce the Discussion, it is much too long… Secondly, please reduce the discussion by removing unnecessarily lengthy comparisons with other literature. For example, discussion on wind direction: State that the following references found flies did orientate with wind direction, while the following did not. You do not need to explain what each reference found in detail. 

- The discussion has been edited to remove multiple lengthy literature comparisons. As the reviewer suggested, we have condensed sections of text in an attempt to avoid discussing each reference in extended depth. Specific sections that were shorten include passages addressing literature on wind direction, flight distances, and comparisons of HR tracking with other ways fly movement might be studied. These reductions have resulted in the removal of approximately one and a half pages of text.

I would also give the whole document another detailed read to remove spelling and grammatical errors, I am not sure I caught them all.

- The entire manuscript has been read over again with several additional errors corrected.

Specific comments

Line 58 – corrected as suggested

Line 59 – corrected as suggested

Line 61 – corrected as suggested

Line 70 – corrected as suggested

Line 71 – trapping networks are broader than just Medfly

Line 97 – corrected as suggested

Line 106 – corrected as suggested

Line 108 – corrected as suggested

Line 109 – corrected as suggested

Line 141 – info added as suggested

Line 170 – corrected as suggested

Lines 202-203 – text added to clarify as suggested

Line 206 – text added as suggested

Line 225 – text added as suggested

Line 237 – text added as suggested

Line 279 – corrected as suggested

Line 323 – corrected as suggested

Line 332 – corrected as suggested

Line 347 – corrected as suggested

Lines 358-359 – sentence removed as suggested

Line 385 – corrected as suggested

Line 427 – corrected as suggested

Line 393 – corrected as suggested

Line 401 – corrected as suggested

Line 405 – corrected as suggested

Line 454 – changed to “short-range”

Line 471 – corrected as suggested

Line 474 – corrected as suggested

Line 509 – corrected as suggested

Line 512 – corrected as suggested and sentence removed as suggested

Line 518 – corrected as suggested

Line 520 – corrected as suggested

Line 522 – CRW is defined earlier in the paper

Line 551 – added full name as suggested

Line 555 – added full name as suggested

Line 559 – added full name as suggested

Lines 561-563, 575-578, 586-588, 605-609 – We are not familiar with a scientific writing style guide that forbids results in figure legends. In fact, we found several style guides that encourage the inclusion of brief results and reporting statistics in figure legends. We believe the legends are not overly long and would enhance the readers understanding of the figure. If the editor feels we should remove these we will do so.

Line 565 – added full name as suggested

Line 569 – added full name as suggested

Line 573 – added full name as suggested

Line 580 – added full name as suggested

Line 584 – added full name as suggested

Line 590 – added full name as suggested

Line 611 – added full name as suggested

---

## [Editor Report · Decision Letter 1]

18 Oct 2022

Harmonic radar tracking of individual melon flies, Zeugodacus cucurbitae, in Hawaii: Determining movement parameters in cage and field settings

PONE-D-22-21998R1

Dear Dr. Siderhurst,

We’re pleased to inform you that your manuscript has been judged scientifically suitable for publication and will be formally accepted for publication once it meets all outstanding technical requirements.

Kind regards,

Ramzi Mansour

Academic Editor

PLOS ONE

---

## [Editor Report · Acceptance letter]

20 Oct 2022

PONE-D-22-21998R1 

Harmonic radar tracking of individual melon flies, *Zeugodacus cucurbitae*, in Hawaii: Determining movement parameters in cage and field settings 

Dear Dr. Siderhurst:

I'm pleased to inform you that your manuscript has been deemed suitable for publication in PLOS ONE. Congratulations! Your manuscript is now with our production department. 

Kind regards, 

on behalf of

Dr. Ramzi Mansour 

Academic Editor

PLOS ONE